# Corneal confocal microscopy demonstrates sensory nerve loss in children with autism spectrum disorder

Adnan Khan[1,2]*, Madeeha Kamal[3], Abdula Alhothi[4], Hoda Gad[1], Marian A, Adan[4], Georgios Ponirakis[1], Ioannis N. Petropoulos[1], Rayaz A. Malik[1]*

1 Research Division, Weill Cornell Medicine-Qatar, Doha, Qatar, 2 Faculty of Health Sciences, Khyber Medical University, Peshawar, Pakistan, 3 Department of Pediatrics, Sidra Medicine, Doha, Qatar, 4 Department of Pediatrics, Hamad General Hospital, Doha, Qatar

* ram2045@qatar-med.cornell.edu (RAM); oneineye@yahoo.com (AK)

**Data Availability Statement:** The data used for statistical analysis in this study is available at (https://figshare.com/articles/dataset/ASD/21836214).

## Abstract

Autism spectrum disorder (ASD) is a developmental disorder characterized by difficulty in communication and interaction with others. Postmortem studies have shown cerebral neuronal loss and neuroimaging studies show neuronal loss in the amygdala, cerebellum and inter-hemispheric regions of the brain. Recent studies have shown altered tactile discrimination and allodynia on the face, mouth, hands and feet and intraepidermal nerve fiber loss in the legs of subjects with ASD. Fifteen children with ASD (age: 12.00 ± 3.55 years) and twenty age-matched healthy controls (age: 12.83 ± 1.91 years) underwent corneal confocal microscopy (CCM) and quantification of corneal nerve fiber morphology. Corneal nerve fibre density (fibers/mm$^2$) (28.61 ± 5.74 vs. 40.42 ± 8.95, $p$ = 0.000), corneal nerve fibre length (mm/mm$^2$) (16.61 ± 3.26 vs. 21.44 ± 4.44, $p$ = 0.001), corneal nerve branch density (branches/mm$^2$) (43.68 ± 22.71 vs. 62.39 ± 21.58, $p$ = 0.018) and corneal nerve fibre tortuosity (0.037 ± 0.023 vs. 0.074 ± 0.017, $p$ = 0.000) were significantly lower and inferior whorl length (mm/mm$^2$) (21.06 ± 6.12 vs. 23.43 ± 3.95, $p$ = 0.255) was comparable in children with ASD compared to controls. CCM identifies central corneal nerve fiber loss in children with ASD. These findings, urge the need for larger longitudinal studies to determine the utility of CCM as an imaging biomarker for neuronal loss in different subtypes of ASD and in relation to disease progression.

## Introduction

Autism Spectrum Disorder (ASD) is a complex and heterogenous neurodevelopmental brain disorder affecting 1–2% of children worldwide [1, 2]. It is characterized by an impairment in social communication and restricted/repetitive behaviour attributed to altered levels of neurotransmitters and neuro-axonal development [3]. Most research has focused on brain-centric mechanisms with little attention to peripheral nerve involvement. However, studies have reported abnormal peripheral sensory responses in multiple domains [4–6] in relation to ASD severity [7]. This is now recognized in the autism diagnostic criteria in the Diagnostic and Statistical Manual of Mental Disorders (DSM)-V, as hyper/hypo reactivity to sensory stimuli [8].

**Funding:** Autism Research Institute (http://dx.doi.org/10.13039/100007332 0). The funders had no role in study design, data collection and analysis, decision to publish, or preparation of the manuscript.

**Competing interests:** NO authors have competing interests

A number of recent studies have shown abnormalities in peripheral nerves in subjects with ASD [9–12] and emerging evidence from animal studies indicate that some aspects of ASD are linked to peripheral sensory deficits [13]. In an autistic mouse model, altered tactile discrimination and allodynia on the face, mouth, and paws was related to defects in peripheral somatosensory neurons [14]. In subjects with autism, certain ASD traits were associated with alterations in mechanoreceptor-targeted affective touch [15]. A reduction in parasympathetic activation [9–11] and retinal nerve fiber layer thickness has also been found in children with ASD [16]. Silva & Schalock [17] found reduced intraepidermal nerves in the lower leg of children with autism compared to healthy controls. More recently, Chien et al. [12] reported a reduction in intraepidermal nerve fiber density and increased thermal thresholds in the leg of 32 adults with autism.

We have pioneered the use of corneal confocal microscopy (CCM) to demonstrate corneal nerve loss in a range of peripheral [18, 19] and central [20–22] neurodegenerative diseases. Furthermore, we and others have shown corneal nerve fiber loss in children with diabetic neuropathy [23–25].

The objective of this study was to use CCM and quantify corneal nerve morphology in children with ASD compared to age-matched controls.

## Materials and methods

This is a cross-sectional, observational study conducted in Doha, Qatar. Fifteen children with ASD and twenty age-matched healthy controls were studied. A senior pediatric consultant in neurodevelopmental disorders (MK) established the diagnosis of ASD according to the DSM-V criteria [8]. Demographic and clinical data were obtained from the patients' health records. Exclusion criteria included premature birth, neuro-psychiatric conditions, attention deficit hyperactivity disorder, isolated intellectual disability, a known history of ocular trauma or surgery, high refractive error, glaucoma, dry eye, and corneal dystrophy [26].

All procedures were in accord with the ethical standards of Weill Cornell Medicine-Qatar (IRB No: 19–00016) and Hamad Medical Corporation (MRC No: MRC-01-20-761) and the 1964 Helsinki declaration and its later amendments. Written informed consent was obtained from the parents or legal guardians, with written assent from all participants.

### Corneal confocal microscopy

All participants underwent CCM (Heidelberg Retinal Tomograph III Rostock Cornea Module; Heidelberg Engineering GmbH, Heidelberg, Germany). To perform examination, a local anesthetic (0.4% benoxinate hydrochloride; Chauvin Pharmaceuticals, Chefaro, United Kingdom) was used to anesthetize both eyes, and Viscotears (Carbomer 980, 0.2%, Novartis, United Kingdom) was used as the coupling agent between the cornea and the cap. The examination took approximately 15 minutes for both eyes. The examiner captured images of the sub-basal nerve plexus using the section mode. Based on depth, contrast, focus, and position, 6 images per participant were selected [27]. All CCM images were analyzed using CCMetrics (M. A. Dabbah, ISBE, University of Manchester, Manchester, United Kingdom). Corneal nerve fiber density (CNFD), corneal nerve branch density (CNBD), corneal nerve fiber length (CNFL), corneal nerve fiber tortuosity (CNFT), and inferior whorl length (IWL, available for seven subjects) were manually analyzed [18].

### Statistical analysis

Prism (version 9.1.0 for Mac, GraphPad Software Inc, San Diego, CA, US), was used for graphic illustrations and IBM SPSS Statistics software (Version 26) was used to perform statistical analyses. Shapiro-Wilk test was used to confirm the normal distribution of the data.

Fisher's exact test was used to calculate differences in categorical data and an unpaired t-test was used for continuous data. Pearson correlation was performed for correlation between CCM and clinical parameters. Data are expressed as mean ± SD and P<0.05 was considered significant. The data used for statistical analysis in this study is available at (https://figshare.com/articles/dataset/ASD/21836214).

## Results

### Clinical and metabolic characteristics

Children with ASD (n = 15) had a comparable age ($p = 0.423$), gender ($p = 0.267$), height ($p = 0.268$), weight ($p = 0.404$) and body mass index (BMI) ($p = 0.769$) compared to controls (n = 20) (Table 1).

### Corneal nerve parameters

CNFD (28.61 ± 5.74 vs. 40.42 ± 8.95, $p = 0.000$), CNFL (16.61 ± 3.26 vs. 21.44 ± 4.44, $p = 0.001$), CNBD (43.68 ± 22.71 vs. 62.39 ± 21.58, $p = 0.018$) and CNFT (0.037 ± 0.023 vs. 0.074 ± 0.017, $p = 0.000$) were significantly lower in children with ASD compared to controls (Figs 1 and 2). There was no significant difference in IWL (21.06 ± 6.12 vs. 23.43 ± 3.95, $p = 0.255$) between subjects with ASD and controls (Table 1).

### Correlation between CCM parameters and clinical and metabolic parameters

Age, height, weight, BMI, systolic and diastolic BP did not correlate with any CCM parameter except for age with CNFT (r = 0.363, $p = 0.032$) (Table 2).

## Discussion

This is the first study to demonstrate reduced corneal nerves in children with ASD compared to healthy controls. Our findings support the emerging thesis that there is peripheral neurodegeneration in children with ASD [12, 17].

**Table 1. Clinical, demographic and CCM measures in children with ASD and controls.**

| Characteristics | Controls | ASD | P Value |
|---|---|---|---|
| Number of Participants | 20 | 15 | |
| Age, years | 12.83 ± 1.91 | 12.00 ± 3.55 | 0.423 |
| Gender (M/F), n | 11/9 | 11/4 | 0.267 |
| Height, meter | 1.45 ± 0.13 | 1.51 ± 0.17 | 0.268 |
| Weight, kg | 47.87 ± 18.63 | 54.25 ± 26.12 | 0.404 |
| BMI, kg/m$^2$ | 22.27 ± 5.47 | 22.91 ± 7.27 | 0.769 |
| Systolic BP, mmHg | NA | 111.46 ± 13.82 | NA |
| Diastolic BP, mmHg | NA | 68.38 ± 10.65 | NA |
| **CCM Parameters** | | | |
| CNFL, mm/mm$^2$ | 21.44 ± 4.44 | 16.61 ± 3.26 | **0.001** |
| CNFD, fibers/mm$^2$ | 40.42 ± 8.95 | 28.61 ± 5.74 | **0.000** |
| CNBD, branches/mm$^2$ | 62.39 ± 21.58 | 43.68 ± 22.71 | **0.018** |
| CNFT, TC | 0.074 ± 0.017 | 0.037 ± 0.023 | **0.000** |
| IWL, mm/mm$^2$ | 23.43 ± 3.95 | 21.06 ± 6.12 | 0.255 |

Results are expressed as mean ± SD. Statistically significant differences between groups using t-test ($p < 0.05$). BMI (Body mass index), BP (blood pressure), CNFL (corneal nerve fiber length), CNFD (corneal nerve fiber density), CNBD (corneal nerve branch density), corneal nerve fiber tortuosity (CNFT), and inferior whorl length (IWL).

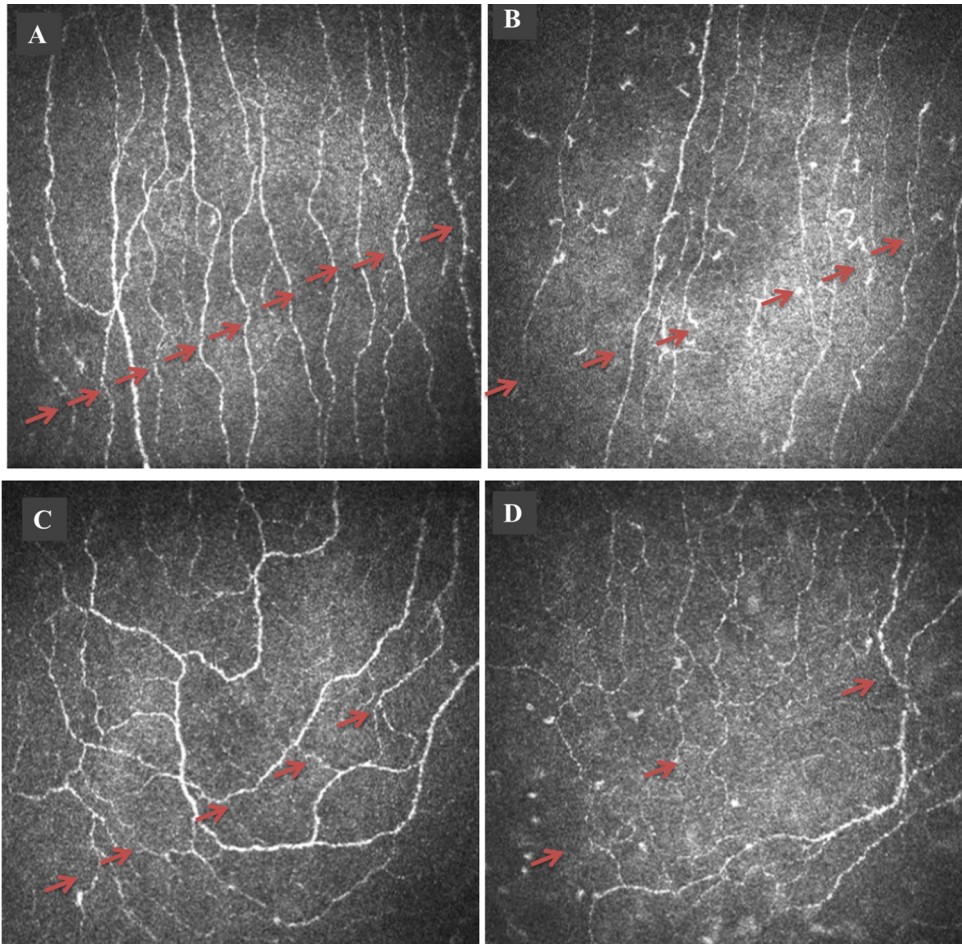

**Fig 1. Central corneal nerve image from a healthy control (A) and child with ASD (B) showing a reduction in main nerve fibers and branches (red arrows) in children with ASD and Inferior whorl (IW) image from a healthy control (C) and child with ASD (D) showing no difference.**

Neuroimaging studies have confirmed altered brain development and neural connectivity and prospective studies have shown altered trajectories for normal brain development preceding the overt presentation of ASD by many years [28, 29]. Several postmortem studies of the brain of patients with autism have shown neuronal loss in the cerebellum and anterior cingulate cortex [30–34], amygdala [35] and fusiform gyrus [36]. Furthermore, there is evidence of primary neurodegeneration of Purkinje cells with gliosis, suggesting that these changes are acquired rather than neurodevelopmental [37]. Gialloreti et al. [16] showed a reduction in retinal nerve fibre layer thickness in children with high functioning autism, which correlated with verbal-IQ/ performance-IQ discrepancy and García-Medina et al. [38] found thicker retinae in children with ASD which correlated with cognitive function.

In the current study, corneal nerve density, length and branch density were lower in children with ASD, independent of anthropometric and clinical risk factors which have previously been associated with corneal nerve loss [39]. Interestingly, corneal nerve branch density was high in 3 individuals with ASD. Previously, we have also shown that CNBD was higher in patients with Parkinson's disease [40] and was related to the perception of affective touch [41]. This may reflect the range of altered sensory phenotypes with both hypo and hyper responsiveness in ASD. We show reduced tortuosity of the corneal nerves in subjects with ASD,

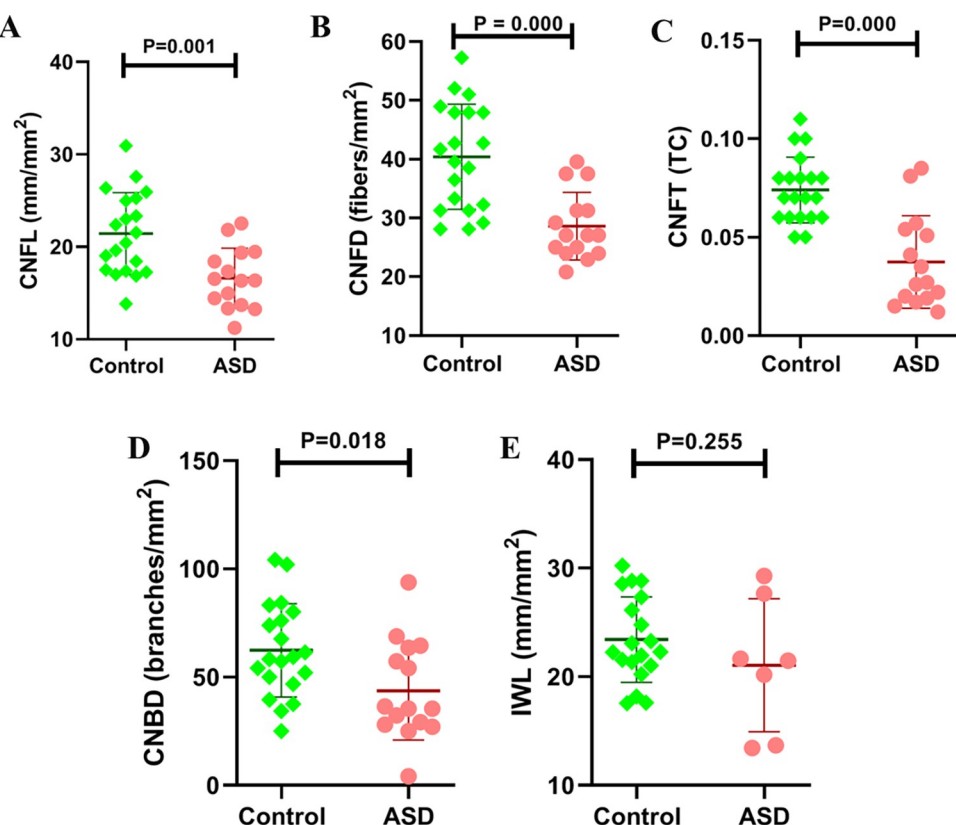

**Fig 2. Corneal nerve fiber length (CNFL) (A), corneal nerve fiber density (CNFD) (B), corneal nerve fiber tortuosity (CNFT) (C), corneal nerve branch density (CNBD) (D) and inferior whorl length (IWL) (E) presented as individual data points in children with ASD (red circles) and controls (green dots) and as the mean and standard deviation.**

**Table 2. Correlations between corneal confocal microscopy parameters and clinical and metabolic parameters.**

| Parameters | Age | Height | Weight | BMI | Systolic BP | Diastolic BP |
|---|---|---|---|---|---|---|
| CNFD | 0.019 | -0.063 | -0.005 | 0.129 | -0.543 | -0.525 |
| r | | | | | | |
| P | 0.915 | 0.722 | 0.979 | 0.468 | 0.055 | 0.066 |
| CNBD | -0.134 | -0.211 | -0.111 | 0.021 | -0.184 | -0.365 |
| r | | | | | | |
| P | 0.443 | 0.23 | 0.527 | 0.907 | 0.547 | 0.221 |
| CNFL | -0.064 | -0.127 | -0.062 | 0.068 | -0.384 | -0.29 |
| r | | | | | | |
| P | 0.715 | 0.475 | 0.724 | 0.702 | 0.195 | 0.336 |
| CNFT | **0.363***| -0.05 | -0.14 | -0.138 | -0.121 | 0.027 |
| r | | | | | | |
| p | **0.032** | 0.777 | 0.422 | 0.435 | 0.694 | 0.93 |
| IW | 0.034 | 0.119 | 0.086 | 0.165 | 0.204 | -0.029 |
| r | | | | | | |
| P | 0.87 | 0.572 | 0.677 | 0.431 | 0.66 | 0.951 |

BMI, body mass index; CNBD, corneal nerve branch density; CNFD, corneal nerve fiber density; CNFL, corneal nerve fiber length; CNFT, corneal nerve fiber tortuosity; IWL, inferior whorl length; TC, tortuosity coefficient; BP, blood pressure.

comparable to children with celiac disease [23], but this contrasts with studies showing increased tortuosity in adults with diabetic neuropathy [42] and no change in children with type 1 diabetes [25]. This reduced nerve fibre tortuosity may represent an alteration in corneal nerve morphology which may be unique to subjects with ASD. We show no loss of corneal nerves at the more distal inferior whorl which may indicate a unique pattern of neurodegeneration affecting proximal rather than distal nerves. This differs from our previous studies in adults with diabetic neuropathy showing greater loss of nerves at the inferior whorl compared to the central cornea, which is consistent with a dying back neuropathy [43]. Previously, Silva & Schalock [17] found reduced intra epidermal nerve fiber density in the lower leg of four children with ASD and hypoesthesia and allodynia. More recently, Chien et al. [12] found reduced intraepidermal nerve fiber density in more than half of 36 adult males with ASD and found a U-shaped relationship with autism severity, possibly reflecting hypo/hyper responses to sensory inputs. Skin biopsy is of course an invasive procedure, which is difficult to perform in children and the quantification of intraepidermal nerve fibers requires considerable expertise. In comparison, CCM is a rapid non-invasive imaging technique used in ophthalmic practice to study the cornea. It requires minimal expertise to capture corneal nerve images, and automated software allows rapid quantification of corneal nerve damage.

Corneal confocal microscopy is an established ophthalmic imaging modality for identifying small nerve fiber loss in diabetic [18] and other neuropathies [44, 45] and predicts the development of diabetic neuropathy [46]. We and others have also shown significant corneal nerve loss in multiple sclerosis [21, 22], Parkinson's disease [47], dementia [20], and stroke [48], which was associated with neurological disability. CCM has also been used to show nerve regeneration in clinical trials of disease-modifying therapies [19, 49].

There are distinct subtypes of autism, but there are no precise ways of identifying these categories and there is a lack of knowledge regarding disease trajectories [50]. A precision medicine strategy is essential for translational research and successful drug development in autism [51, 52]. In 2016 the European Medicines Agency (EMA) endorsed 5 major outcomes to enable stratification of people with ASD: 1) EEG to measure deficits in social cognition; 2) MRI; 3) Eye tracking; 4) Measures of executive function and basic emotions and 5) Methods to identify clinical outcome. Although, eye tracking, fMRI and diffusion tensor imaging (DTI) show promise they cannot accurately identify precise patient endophenotypes, or monitor progression or recovery in ASD [53].

We acknowledge this is an exploratory study with a small sample size limiting the generalizability of our findings. Furthermore, CCM needs cooperative participants and therefore its utility may be limited to children with high functioning ASD.

Nevertheless, we believe CCM could act as a rapid, non-invasive surrogate imaging biomarker of neurodegeneration in ASD. Larger studies are warranted to determine the relationship between corneal nerve loss and changes observed using brain neuroimaging as well as behavioral alterations in patients with ASD.

## Author Contributions

**Conceptualization:** Adnan Khan.

**Data curation:** Adnan Khan, Hoda Gad, Marian A, Adan, Georgios Ponirakis, Ioannis N. Petropoulos.

**Formal analysis:** Adnan Khan.

**Funding acquisition:** Adnan Khan, Rayaz A. Malik.

**Investigation:** Adnan Khan, Madeeha Kamal.

**Methodology:** Adnan Khan.

**Project administration:** Abdula Alhothi, Rayaz A. Malik.

**Resources:** Abdula Alhothi.

**Software:** Adnan Khan.

**Supervision:** Rayaz A. Malik.

**Visualization:** Rayaz A. Malik.

**Writing – original draft:** Adnan Khan.

**Writing – review & editing:** Adnan Khan, Rayaz A. Malik.

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
