## [Decision Letter · Decision Letter 0]

23 May 2023

PONE-D-23-04083Corneal confocal microscopy demonstrates sensory nerve loss in children with autism spectrum disorderPLOS ONE

Dear Dr. Malik,

Thank you for submitting your manuscript to PLOS ONE. After careful consideration, we feel that it has merit but does not fully meet PLOS ONE’s publication criteria as it currently stands. Therefore, we invite you to submit a revised version of the manuscript that addresses the points raised during the review process.

The manuscript was evaluated by two reviewers.There are several minor revisions existed in the present form. 

See the suggestions carefully, and respond them appropriately.

We look forward to receiving your revised manuscript.

Kind regards,

Masaki Mogi

Academic Editor

PLOS ONE

- https://doi.org/10.1002/ana.26484

- https://doi.org/10.1161/STROKEAHA.117.018289

In your revision ensure you cite all your sources (including your own works), and quote or rephrase any duplicated text outside the methods section. Further consideration is dependent on these concerns being addressed.

“NO authors have competing interests”

6. Please note that in order to use the direct billing option the corresponding author must be affiliated with the chosen institute. Please either amend your manuscript to change the affiliation or corresponding author, or email us at plosone@plos.org with a request to remove this option.

Reviewers' comments:

Reviewer's Responses to Questions

**Comments to the Author**

1. Is the manuscript technically sound, and do the data support the conclusions?

Reviewer #1: Yes

Reviewer #2: Yes

2. Has the statistical analysis been performed appropriately and rigorously? 

Reviewer #1: Yes

Reviewer #2: Yes

3. Have the authors made all data underlying the findings in their manuscript fully available?

Reviewer #1: Yes

Reviewer #2: Yes

4. Is the manuscript presented in an intelligible fashion and written in standard English?

Reviewer #1: Yes

Reviewer #2: Yes

5. Review Comments to the Author

Reviewer #1: Summary

Overall, the paper is well and clearly written. This is an interesting study where authors for the first time demonstrate corneal nerve fibre loss in children with autism spectrum disorder (ASD).

Questions

1. It’s interesting that no reduction in IWL was observed in children with ADS. Perhaps the reason for this could be the fact that only 6 children had IW captured.

2. Were the IW images captured for 6 or 7 children? Figure 2E seems to show that there where 7 children with captured IW.

3. Have you considered to quantify Langerhans cells for the same images? This would be interesting to see if there are any differences between both groups.

4. Perhaps it would be beneficial for readers to add IW images as well even there was no significant difference between both groups.

Reviewer #2: The authors have demonstrated a reduction in corneal nerve fibers in children with autism spectrum disorder. This paper represents an important contribution to the literature since this is the first report of corneal nerve loss in ASD. The paper is very well written and the methodology is sound.

I have two minor comments:

1. In the Methods section (page 4, line 24), the authors stated that inferior whorl length parameter was available for six participants, which I think should have been seven – as it appears in Figure 2E.

2. Page 9, line 20; please edit as “… an alteration”

6. PLOS authors have the option to publish the peer review history of their article (what does this mean?). If published, this will include your full peer review and any attached files.

Reviewer #1: No

Reviewer #2: No

---

## [Author Response · Author response to Decision Letter 0]

8 Jun 2023

Reviewers

Reviewer:#1: Summary

Overall, the paper is well and clearly written. This is an interesting study where authors for the first time demonstrate corneal nerve fibre loss in children with autism spectrum disorder (ASD).

Questions

It is interesting that no reduction in IWL was observed in children with ASD. One possible reason for this could be the fact that only 6 children had IW captured. 

Answer: Thank You. We agree that with IW data from only 7 individuals may well have limited the power of the study. An alternate explanation is that lack of change in the IW reflects less distal nerve involvement and therefore provides insights into the underlying pathology of neurodegeneration in ASD. This differs from greater IW involvement in patients with diabetic neuropathy who typically have a distal dying back neuropathy. We have now expanded the discussion to clarify this on page 9.

“We show no loss of corneal nerves at the more distal inferior whorl which may indicate a unique pattern of neurodegeneration affecting proximal rather than distal nerves. This differs from our previous studies in adults with diabetic neuropathy showing greater loss of nerves at the inferior whorl compared to the central cornea, which is consistent with a dying back neuropathy [43].” 

Were the IW images captured for 6 or 7 children? Figure 2E seems to show that there were 7 children with captured IW. 

Answer: Thank you. We apologize for this error. The inferior whorl length was assessed in seven participants as per the figure. We have now corrected this in the text.

Have you considered quantifying Langerhans cells for the same images? It would be interesting to see if there are any differences between the two groups. 

Answer: We agree that quantification of LC’s may provide insights into the underlying mechanisms of neurodegeneration in ASD. However, the variability in LC assessment if very large and with the small numbers we do not feel we would generate meaningful results. We are planning a larger study where we will quantify LC’s.

It might be beneficial for readers to add IW images as well, even if there was no significant difference between the two groups. 

Answer: Thank you for your suggestion. We have now added IW images in Fig 1.

“Fig 1. Central corneal nerve image from a healthy control (A) and child with ASD (B) showing a reduction in main nerve fibers and branches (red arrows) in children with ASD and Inferior whorl (IW) image from a healthy control (C) and child with ASD (D) showing no difference.

Reviewer #2: The authors have demonstrated a reduction in corneal nerve fibers in children with autism spectrum disorder. This paper represents an important contribution to the literature as it is the first report of corneal nerve loss in ASD. The paper is very well-written, and the methodology is sound.

I have two minor comments: In the Methods section (page 4, line 24), the authors stated that the inferior whorl length parameter was available for six participants, which I believe should have been seven, as it appears in Figure 2E. 

Answer: Thank you. We apologize for this error. The inferior whorl length was assessed in seven participants as per the figure 2E. We have now corrected this in the text.

Page 9, line 20; please edit as "... an alteration." 

Answer: Thank you. We have edited the sentence accordingly.

---

## [Decision Letter · Decision Letter 1]

26 Jun 2023

Corneal confocal microscopy demonstrates sensory nerve loss in children with autism spectrum disorder

PONE-D-23-04083R1

Dear Dr. Malik,

We’re pleased to inform you that your manuscript has been judged scientifically suitable for publication and will be formally accepted for publication once it meets all outstanding technical requirements.

Kind regards,

Masaki Mogi

Academic Editor

PLOS ONE

Additional Editor Comments (optional):

The authors well responded to the Reviewers' suggestions. No further comment.

Reviewers' comments:

Reviewer's Responses to Questions

**Comments to the Author**

1. If the authors have adequately addressed your comments raised in a previous round of review and you feel that this manuscript is now acceptable for publication, you may indicate that here to bypass the “Comments to the Author” section, enter your conflict of interest statement in the “Confidential to Editor” section, and submit your "Accept" recommendation.

Reviewer #2: All comments have been addressed

2. Is the manuscript technically sound, and do the data support the conclusions?

Reviewer #2: (No Response)

3. Has the statistical analysis been performed appropriately and rigorously? 

Reviewer #2: (No Response)

4. Have the authors made all data underlying the findings in their manuscript fully available?

Reviewer #2: (No Response)

5. Is the manuscript presented in an intelligible fashion and written in standard English?

Reviewer #2: (No Response)

6. Review Comments to the Author

Reviewer #2: (No Response)

7. PLOS authors have the option to publish the peer review history of their article (what does this mean?). If published, this will include your full peer review and any attached files.

Reviewer #2: No

---

## [Editor Report · Acceptance letter]

4 Jul 2023

PONE-D-23-04083R1 

Corneal confocal microscopy demonstrates sensory nerve loss in children with autism spectrum disorder 

Dear Dr. Malik:

I'm pleased to inform you that your manuscript has been deemed suitable for publication in PLOS ONE. Congratulations! Your manuscript is now with our production department. 

Kind regards, 

on behalf of

Dr. Masaki Mogi 

Academic Editor

PLOS ONE